# Data-Mining Poultry Processing Bio-Mapping Counts of Pathogens and Indicator Organisms for Food Safety Management Decision Making

**DOI:** 10.3390/foods12040898

**Published:** 2023-02-20

**Authors:** David A. Vargas, Juan F. De Villena, Valeria Larios, Rossy Bueno López, Daniela R. Chávez-Velado, Diego E. Casas, Reagan L. Jiménez, Sabrina E. Blandon, Marcos X. Sanchez-Plata

**Affiliations:** International Center for Food Industry Excellence, Department of Animal and Food Sciences, Texas Tech University, Lubbock, TX 79409, USA

**Keywords:** data mining, poultry processing shift analysis, kernel density estimation, poultry pathogen–indicator relationship

## Abstract

Bio-mapping studies play an important role, as the data collected can be managed and analyzed in multiple ways to look at process trends, find explanations about the effect of process changes, activate a root cause analysis for events, and even compile performance data to demonstrate to inspection authorities or auditors the effect of certain decisions made on a daily basis and their effects over time in commercial settings not only from the food safety perspective but also from the production side. This study presents an alternative analysis of bio-mapping data collected throughout several months in a commercial poultry processing operation as described in the article “Bio-Mapping Indicators and Pathogen Loads in a Commercial Broiler Processing Facility Operating with High and Low Antimicrobial Interventions”. The conducted analysis identifies the processing shift effect on microbial loads, attempts to find correlation between microbial indicators data and pathogens loads, and identifies novel visualization approaches and conducts distribution analysis for microbial indicators and pathogens in a commercial poultry processing facility. From the data analyzed, a greater number of locations were statistically different between shifts under reduced levels of chemical interventions with higher means at the second shift for both indicators and pathogens levels. Minimal to negligible correlation was found when comparing aerobic counts and Enterobacteriaceae counts with *Salmonella* levels, with significant variability between sampling locations. Distribution analysis and visualization as a bio-map of the process resulted in a clear bimodality in reduced chemical conditions for multiple locations mostly explained by shift effect. The development and use of bio-mapping data, including proper data visualization, improves the tools needed for ongoing decision making in food safety systems.

## 1. Introduction

The United States ranks among the largest and most efficient poultry producer in the world and is considered highly competitive in global export markets [1,2]. The total value of production from broilers, eggs, turkeys, and the value of sales from chickens in 2020 was USD 35.5 billion, down 11 percent from USD 40.0 billion in 2019 [2,3]. Poultry processing operations continuously seek to increase efficiency at all locations of the production process, including disease control, breeding, feed compositions, and housing systems at grow-out facilities [1]. High production levels are in tune with the large volumes of poultry meat consumption (broilers, other chicken, and turkey) in the U.S., which are considerably higher than beef or pork counterparts [4]. Poultry consumption is incentivized by becoming the lowest-priced meat, with the average per capita consumption of chicken in the 95.6 pounds range annually [5].

The Center for Disease Control and Prevention (CDC) estimates that each year in the United States, over 47.8 million people get sick, 127,839 people require hospitalization, and 3037 die from foodborne diseases [6]. The World Health Organization estimates that over 2 million people die each year from diarrheal diseases mainly caused by the ingestion of contaminated foods [7]. Processing facilities remain concerned and actively search for alternatives to control commonly originated pathogens associated with raw chicken, such as *Salmonella* and *Campylobacter* spp., that can lead to foodborne illnesses caused by the consumption of poultry meat products [8]. *Salmonella* causes about 1.35 million infections, 26,500 hospitalizations, and 420 deaths in the U.S. every year [9]. The CDC indicates that *Campylobacter* affects around 1.5 million U.S. residents every year, and most cases are not part of recognized outbreaks due to underreporting [10]. Despite the fact that both of these microorganisms have been attributed to foodborne outbreaks across multiple food categories [11], poultry meat remains one of the main targets of regulators and consumers as responsible for illness from these two organisms, and efforts to reduce their contribution are ongoing. To reduce consumer risk, microbial interventions are used in both the pre-harvest and post-harvest production environments [8]. However, many poultry processors continue to have post-intervention samples test positive, and compliance with the U.S. Department of Agriculture (USDA) performance standards is an ongoing challenge [12,13].

Peroxyacetic acid (PAA) has become the leading choice in processing plants as an antimicrobial applied in several carcass rinse locations, in pre-chiller and chiller applications, as well as post-chill immersion tanks; and the results of a recent study demonstrated that PAA was the most effective antimicrobial currently in use in commercial settings [14]. Cross-contamination throughout the processing chain influences the contamination levels of the entire production flock that is processed in the same line [15]. A 2011 study uncovered a source of cross-contamination during the defeathering step, which concluded that *Campylobacter* is transferred from an individual carcass to another [15]. *Campylobacter* is one of the major causes of bacterial food-borne diarrheal diseases worldwide and can be carried in the intestines, liver, and other organs of animals [7,10]. A previous study showed that 43 of 680 samples from a cleaned and disinfected slaughter process, 70 of 300 neck skin samples after chilling and 24 of 240 thigh samples were *Salmonella*-positive when all flocks first had an initial *Salmonella*-negative status [16]. In any case, most processors evaluate the hygienic performance of antimicrobial intervention schemes by collecting samples at various stages of processing on an ongoing basis and analyzing for microbial indicators and, in some instances, pathogen prevalence.

Poultry processors seek new technologies to assess process hygienic performance to demonstrate compliance with performance standards. Rapid detection methods for prevalence and recent developments in pathogen quantification are important and needed for timely decision making in food safety management systems. These enumeration methods gather information that helps and allows processors to potentially know the levels of contamination such as that of *Salmonella* and *Campylobacter* spp. of each lot that enters the processing line. It also facilitates the responsibilities of technical managers to focus on the mitigation of food safety hazards in any way possible and to prevent other flocks from becoming contaminated in the first place, especially from sampling locations with greater microbial load. The areas of hanging, scalding, and plucking have consistently been identified as the most contaminated sampling zones before commencement of slaughter [16]. Conventional microbiology and molecular methods have been used to identify differences between shifts and how the levels of contamination of one flock can influence subsequent flocks during processing [17]. Risk-based slaughter programming is expected to help minimize the likelihood of cross-contamination from a *Salmonella*-positive to a *Salmonella*-negative flock, but it depends on the pathogen loads of previous flocks [16]. The availability of feasible pathogen quantification methods would be more valuable to determine the efficacy of process control interventions, corrective actions, and final product microbial performance to make rapid, within-shift food safety decisions [12]. 

The USDA’s Food Safety and Inspection Service (FSIS) is responsible for the enforcement of the Poultry Products Inspection Act. FSIS conducted the Raw Chicken Parts Baseline Survey (RCPBS) to compare the percent of positive tests and levels of *Salmonella* and *Campylobacter* as well as the levels of generic *Escherichia coli*, aerobic plate count (APC), Enterobacteriaceae, and total coliforms as microbial indicators to determine if significant differences existed between processing steps and production shifts. However, this study did not show any statistically significant difference for any pathogen or indicator bacteria for the percent of positive samples or levels of bacteria target between shifts (*p* > 0.05) [18]. In addition, the FSIS has overseen the modernization of poultry processing inspection systems, and that entails the use of ongoing and comprehensive microbial data surveillance to demonstrate process control and performance standards compliance. 

Poultry processors have assimilated modernization components, and the whole industry is moving towards a modern inspection system that includes large datasets generation that requires ongoing analysis and management for proper utilization in decision-making activities. Since changing from traditional inspection systems where minimal data was collected, chicken processing facilities are becoming more and more skilled at collecting data and utilizing this information in food safety management systems. While data science is a powerful tool that generates significant value on collected datasets and allows companies to have better knowledge and understanding of what is happening in the process on a regular basis, the time and skills to capture the value of data science are currently in the developmental stages, and the use of data analysis and visualization for decision making and to find solutions to microbial control operations is somehow limited [19]. 

Bio-mapping studies play an important role, as the data collected and the ongoing data collected through monitoring systems can be managed and analyzed in multiple ways to look at process trends, find explanations about process changes, trigger root cause analysis for specific microbial events, and even compile microbial performance data to demonstrate the effect of certain decisions made on a daily basis and their effects over time to inspection authorities or auditors in commercial settings. The overall goal of this study was to show different perspectives and approaches for data analysis by taking advantage of a comprehensive bio-mapping study compiled in a prior research project and to demonstrate how this analysis can be used by the processor for making decisions that are sound, robust, and supported by comprehensive information of the process.

## 2. Materials and Methods

### 2.1. Sample Collection, Indicator Enumeration, and Pathogen Detection and Quantification

This study presents an alternative analysis of bio-mapping data collected in a commercial poultry processing operation as described in the article “Bio-Mapping Indicators and Pathogen Loads in a Commercial Broiler Processing Facility Operating with High and Low Antimicrobial Interventions” by De Villena et al. (2022) [2]. Consequently, all the methodology for sample collection, antimicrobial intervention schemes, indicator enumeration, and pathogen detection and quantification are described in detail in the referenced publication. Briefly, whole birds and parts rinses were collected at ten different locations throughout the slaughtering, evisceration, and deboning process (live receiving, rehanger, post-eviscerator, post-cropper, post-neck-breaker (Post-NB), post-inside-outside bird washer 1 (Post-IOBW#1), post-inside-outside bird washer 2 (Post-IOBW#2), pre-chilling, post-chilling, and parts (Wings)) under normal and reduced chemical intervention levels for a total period of 25 months. Microbial indicators (aerobic counts and Enterobacteriaceae) and pathogens (*Campylobacter* and *Salmonella*) were enumerated using the TEMPO^®^ system (BioMérieux, Paris, France) and BAX^®^-System-SalQuant^®^ (Hygiena, Camarillo, CA, USA). No additional sampling or microbial analysis was performed for the preparation of this article, and all the raw data are exactly the same as used for De Villena’s et al., 2022 publication. However, the data mining and statistical analysis approach utilized for trend analysis and visualization of the data takes advantage of additional tools to extract actionable information for food safety management decision making.

### 2.2. Statistical Analysis

All data were analyzed using R (Version 4.1.3) statistical analysis software, and all microbial counts (indicators and pathogens) were transformed to Log CFU/mL of rinse with the exception of *Salmonella* counts, which were transformed to Log CFU/sample (Log CFU/400 mL) due to low quantification levels and to enhance data visualization interpretation. A *p*-value of 0.05 or less was used to determine statistically significant differences.

#### 2.2.1. Shift Comparison

During a 25-month sampling period of operation, 1309 samples were collected, including whole chicken carcass rinses and composite parts rinses, interchangeably during two consecutives 8 h shifts (first and second shift) on a commercial poultry processing facility to account for flock-to-flock variation and day-to-day process variability. After finishing the second shift, the plant has a third shift where the facility is deeply cleaned and sanitized. A *t*-test was performed to compare the counts’ differentiation between shifts at each sample location for normal chemical intervention levels (CX) and reduced chemical levels (RC). If parametric assumptions were not met, the Wilcoxon sum rank test or Mann–Whitney test was used as a non-parametric alternative to determine statistically significant differences. 

#### 2.2.2. Indicators vs. Pathogens Correlation

During the experiment, all samples collected from the processing line were labeled in a way that allowed for the enumeration of indicator microorganisms (aerobic counts and Enterobacteriaceae) as well as pathogen bacterial levels (*Salmonella* and *Campylobacter*), thus allowing us to compare the indicator vs. pathogen levels from the same collected sample. For the correlation analysis, all samples that were not quantifiable for *Salmonella* were removed from consideration, and Enterobacteriaceae and aerobic counts were transformed to Log CFU/sample (Log CFU/400 mL), as was done for the pathogen levels, resulting in 370 samples. Moreover, a new grouping of sampling locations was done according to the similarity of the processing step and the lack of statistically significant differences found between sample locations (incoming = live receiving; feather removal = rehanger; viscera removal = post-eviscerator, post-cropper, and post-neck-breaker; carcass wash = Post-IOBW #1, Post-IOBW #2, and pre-chilling). The post-chilling and wings sampling points were removed because of the very low prevalence and counts obtained during the duration of the bio-mapping experiment. A Pearson correlation analysis was performed by comparing counts for aerobic counts and Enterobacteriaceae with *Salmonella*.

#### 2.2.3. Indicator and Pathogen Distribution

Microbial level distribution plots were generated using the “geom_density_ridges” function from the *ggridges* package in R. The function computes and draws kernel density estimates, which is a non-parametric approach to estimate the probability density function of a continuous variable, in this case, microbial counts of indicators and pathogen levels. The bandwidth was automatically calculated and provided by the function, and the scaling factor was set to 1, indicating that the maximum point of any ridgeline touches the baseline right above. Distribution plots were generated for indicator and pathogen microorganisms for normal and reduced chemical interventions. Furthermore, distribution tables were generated by converting the continuous variable “Log CFU/mL or sample” to a discrete variable expressed as percentage of counts falling within a specific range for all indicator and pathogen microorganisms analyzed in this study.

## 3. Results and Discussion

The collection of a large dataset of microbial indicator and pathogenic levels on the same collected samples during a 25-month surveillance period created the opportunity to analyze different process performance parameters including shift effects, correlation between microbial estimations and sample combinations, pathogen/indicators, and data distribution levels, among others. A detailed description about concentrations and type of interventions about the experiment can be found in De Villena’s et al., 2022 publication [2]. Briefly, the normal processing conditions included chemical interventions such as peroxyacetic acid (PAA), PAA + sodium hydroxide, and sodium hypochlorite at several steps in the evisceration, chilling, and deboning processes, respectively. The reduced chemical processing conditions removed all chemical interventions from various locations except for where needed as per the validated hazard analysis and critical control points (HACCP) plan validated by the FSIS [2]. 

Differences between intervention conditions (normal vs. reduced) for all indicators and pathogens throughout the processing line are described in De Villena’s et al., 2022 publication [2]. However, a summary is as follows:(a)The use of pathogen quantification can improve the use of risk assessment where interventions can target specific stages with higher loads of indicator and pathogen bacteria.(b)Non-difference between normal and reduced in chemical interventions in certain locations suggests the application of chemical interventions in strategic locations.(c)The use of prevalence as a sole measurement of food safety performance can lead to inadequate results.

### 3.1. Shift Comparison

For shift effect assessments, aerobic counts and Enterobacteriaceae counts were evaluated under normal and reduced chemical conditions throughout the whole chicken slaughter process. However, for visualization purposes, only Enterobacteriaceae counts are presented in this paper in Figure 1, as both indicator bacteria showed similar trends during the analysis. Moreover, the location “live receiving” was removed from Figure 1 and Figure 2 because at that sampling location, there is no intervention applied on the birds, so there are no differences between normal and reduced chemical intervention groups.

Enterobacteriaceae counts were significantly different (*p* < 0.05) between shifts under normal chemical conditions at Post-IOBW #1 and Post-IOBW#2 locations, while under reduced chemical conditions differences were found at rehanger, post-eviscerator, post-cropper, post-neck breaker, and Post-IOBW #2. In all locations, for both microbial intervention scheme conditions, the average counts were higher on the second shift when compared with the first one, and those differences are more evident under reduced chemical conditions. These results support the expected and inherent differences between shifts in most food processing plants, where there is accumulation of bacteria throughout the production day that can be responsible for an increase on the risk of cross-contamination between carcasses and surfaces over the day of operation. Moreover, the effectiveness of the post-operational sanitation process can be shown where, even though at the end of the day there is a significant increase in the number of bacteria accumulated during the processing day, a third shift focused on plant sanitization reduces those levels significantly. Data such as this can be used to verify sanitation programs, adjust performance, and compare sanitation systems for an operator. 

There are some locations, such as post-chilling, where even under reduced chemical conditions, no difference was observed between shifts (*p* = 0.12). The chilling step appears to significantly overcome potential shift differences due to the nature of the rinsing effects and chemical performance of the interventions applied (e.g., temperatures below 4 °C, constant mechanical and rinsing action, and PAA at a set concentration (15–100 ppm)), aimed at minimizing the accumulation of bacteria in the chilling system. Minimal numeric differences can be explained by the variable incoming microbial loads of chicken carcasses throughout the processing day, and this can be seen in the size of the boxplot figures.

Pre- and post-operational cleaning and sanitation procedures play a critical role in food safety systems for commercial operations by protecting food from continuous contamination with pathogenic microorganisms from equipment, surfaces, or workers; however, evidence to demonstrate these effects in commercial settings with the effects of high and low levels of antimicrobial interventions in the process is limited [20,21,22]. It has been demonstrated that regular cleaning and disinfection is associated with major reductions in pathogens responsible for food poisoning, and it is crucial that staff is properly trained for conducting this activity [23,24,25]. Pre-operational procedures are sanitation activities performed before production begins, while operational procedures are activities conducted during production to keep equipment and surfaces as sanitary as possible to prevent contamination of chicken carcasses [26]. Differences in chemical concentration levels in the intervention locations clearly affect operational procedures in this study, as more locations in the process were found different under reduced chemical conditions when compared with normal chemical conditions. Even though significant statistical differences were noted among pathogen and indicator organisms between shifts, it is evident that operational and food safety consistency between shifts plays an important role in controlling bacterial growth.

Pathogenic organisms associated with poultry meat, specifically *Salmonella* and *Campylobacter,* were also enumerated under normal and reduced chemical conditions throughout the whole chicken slaughter process for shift comparison analysis. Figure 2 shows the results for *Campylobacter* counts and prevalence for both intervention treatments. Even though there are no statistically significant differences throughout all the processing locations, when they exist, the first shift always has the lower concentration of *Campylobacter,* following similar trends as indicator microorganisms. On the other hand, the size of the boxplots provides an estimate of the natural variability that exists in the samples collected, and this variability is in general greater in boxplots for the first shift during normal chemical conditions. These results suggest that during the first shift, the variability observed in the samples is due to the irregularity of *Campylobacter* loads in incoming chicken carcasses, but during the day, and due to the accumulation effect, the cross-contamination with the chicken carcasses reduces the variability of this pathogen load and increases the counts overall [2,27,28]. This trend is also evident when analyzing the prevalence data, as second-shift prevalence results are kept steady at high levels of contamination during the first steps of production.

Furthermore, at the beginning of the shift, specifically during the first part of the production shift (first shift), antimicrobial intervention levels, whether chemical or non-chemical, are being adjusted more frequently as the first birds arrive to the plant for processing. Depending on the performance of the operation, these parameters and/or concentrations are adjusted until an equilibrium is achieved during the operations. Since there is an effort to consider pre-harvest loads of pathogens as decision-making parameters for customized processing, this type of data can be used to make such decisions.

### 3.2. Indicator vs. Pathogen Levels Correlation

The food processing industry uses estimations of microbial indicator organisms to assess microbial control performance when pathogenic data are scarce or difficult to collect in commercial settings [29]. The goal is to understand the dynamics of indicator organisms levels at different processing steps to assume that pathogens loads will follow in similar fashion to closely related indicator organisms [30,31]. However, evidence to show that microbial indicator organisms data can be used to understand pathogen dynamics is limited and at times counter argumentative. It is common to have high indicator organisms counts in pathogen-negative samples and low indicator organisms levels in pathogen-positive samples. The inclusion of pathogen quantitative data allows for a better indicator organism-to-pathogen comparison. For this study, Pearson’s correlation analysis between indicator organisms and pathogen levels was performed at four grouped sampling locations, as shown in Figure 3. Correlation analysis showed that relationship between indicator organisms and *Salmonella* exists in some areas, while in others, it does not, and when it does, the strength of the correlation is usually low. Significant correlation coefficients were obtained for aerobic counts and *Salmonella* at viscera removal and carcass wash locations and for Enterobacteriaceae at incoming stages and viscera removal locations (*p* < 0.05). Significant correlation coefficients for aerobic counts were 0.25 and 0.36 for viscera removal and carcass locations, respectively, and for Enterobacteriaceae were 0.38 and 0.32 for incoming and viscera removal locations, respectively.

The purpose of indicator organisms testing is to utilize a group of microorganisms to identify trends in a food product, and the type of indicator organism is usually focused in having a sentinel measuring method to understand the dynamics of specific foodborne pathogens [32]. For example, *E. coli* indicator quantification looks at fecal coliforms, with an emphasis on enteric pathogens such as *E. coli* [33], whereas Enterobacteriaceae quantification has been used as a potential indicator of *Salmonella* levels, with corresponding limitations in this approach [33]. The low correlation coefficient obtained in this study suggests that risk assessment on a final product should not rely solely on indicator organisms enumeration and pathogen prevalence but rather on pathogen quantification. Nonetheless, Enterobacteriaceae and aerobic plate counts can be done for surveillance and statistical process control analysis as well as for trend analysis for out-of-specification evaluations or to identify upticks in microbial contamination caused by unusual patterns [34]. The combination of pathogen quantification for risk assessment and risk-driven decision making on top of trend analysis with indicator microorganisms can yield more robust datasets for enhanced food safety management decisions.

For analysis purposes, it is important to mention that all samples found negative for *Salmonella* or samples that were not quantifiable but found positive for prevalence using the respective methodology were removed from the analysis. Indicator microorganism count is not enough for accurate decision making on pathogen levels. In addition, due to the increased interest in *Campylobacter* performance standards (not yet enforceable by United States Department of Agriculture–Food Safety and Inspection Service), research has been conducted to identify a good indicator for *Campylobacter* contamination with some successful but very variable results [35]. Enterobacteriaceae and *Escherichia coli* have both been evaluated as potential *Campylobacter* indicators, but the correlation is not consistent at different levels of the indicator, where high concentration of *Escherichia coli* may indicate high *Campylobacter* concentration, but low levels of *Escherichia coli* do not necessarily indicate low levels of *Campylobacter* [35]. The arrival of novel pathogen quantification methodologies for both *Salmonella* and *Campylobacter* provides opportunities to improve this type of needed analysis and will better support decision making in poultry processing operations.

As poultry processors develop risk assessment strategies to improve food safety performance, the need to have a holistic approach to pathogen data analysis is high. Such an approach must include not only indicator organisms testing but also development of statistical process control (SPC) models to further predict the microbial performance of flocks coming into the processing facilities to minimize the food safety impact. The series of microbial results that combine indicator data and pathogen quantification must be analyzed in tandem to develop in-plant performance standards that are not solely based on pathogen prevalence, as is currently being the case.

### 3.3. Indicator and Pathogen Distribution

For chicken rinses and parts throughout the whole process, 98.5% and 99.9% of the mesophilic aerobic count samples were above the limit of quantification (LOQ) (1 CFU/mL) under normal and reduced chemical interventions, respectively, while 99.7% and 99.9% were above LOQ for Enterobacteriaceae under normal and reduced chemical interventions, respectively (Appendix A, Appendix B, Appendix C and Appendix D, Table A1, Table A2, Table A3 and Table A4). Tables for the distribution of pathogenic microorganism levels were assembled for *Salmonella* and *Campylobacter* (Appendix E, Appendix F, Appendix G and Appendix H, Table A5, Table A6, Table A7 and Table A8). These distributions are presented in ranges of factors of 10. The distribution levels of microbial indicators and pathogens in commercial samples can vary due to a number of reasons that include processing location, type of sample matrix, shift, and application of chemical or physical interventions, among others. Kernel density estimation for mesophilic aerobic counts for both normal and reduced chemical interventions at all locations in a poultry processing plant were developed for distribution analysis, as displayed in Figure 4.

Under reduced chemical conditions, a clear bimodality in the distribution can be seen, especially in the initial processing locations of the slaughtering process evaluated in this study. Further analysis was performed to understand the cause of the bimodality during the distribution, and shift averages were plotted on top of the data distribution plots, which perfectly match the two peaks of the bimodal kernel density estimation. If the datasets are not analyzed as a distribution, the compilation of shift and other potential variability effects may hinder proper analysis since the wide variability range will reduce the likelihood of identifying evident patterns that can be hidden behind large datasets of variable distribution caused by specific parameters. This type of analysis provides strong support for clearly identifying all parameters associated with a given sample, such as lot, shift, and time of collection, among others, so that proper data mining can be conducted, adding additional value to bio-mapping data studies.

Based on the kernel density analysis, reduced chemical interventions appear to have a wider distribution when compared to the normal chemical intervention levels. This may be explained by the evident shift differences observed, which closely relates with the reduction of chemical concentrations and their effect at keeping microbial loads low in comparison to normal chemical interventions, which in turn show a narrower distribution due to the ongoing antimicrobial activity despite the initial incoming loads. By the continuous use of chemical interventions, there is a steady concentration effect of bacteria throughout both shifts, as can be observed by the closeness of the averages plotted in the graph under the red distribution bell, potentially reducing the possibility of microbial accumulation and overall risk of product cross contamination. In the final processing steps (post-chilling and wings), maintaining the intervention parameters turns out to be important as the effectiveness of the intervention schemes in reducing bacterial concentration to similar end microbial loads detected despite the higher concentration found in the second shift.

The same kernel distribution plot was made for *Campylobacter* counts and is displayed in Figure 5. Similar microbial patterns as observed in the aerobic count data were seen for this pathogen. The distribution patterns in early processing locations contrast sharply with the more pronounced narrow distribution of *Campylobacter* in the post-chill and parts locations. The concentration of *Campylobacter* in the pre-chilling location varied narrowly, with less than 1 Log CFU/mL range, creating a “cone” effect on the data distribution, which eliminates the shift effect seen in early processing locations. Similar effects can be seen by the use of interventions in the IOBW stations. For wing samples, the distribution comparison of normal chemical vs. reduced chemical shows that the use of antimicrobial interventions helps standardize the pathogen levels and reduce variability, thus supporting their use as process control systems in support of food safety management.

The distribution charts for *Campylobacter* help explain the relative inefficiency of some of the antimicrobial interventions applied at different processing locations when analyzing performance based on prevalence only, as detailed in Figure 2 [2]. Despite clearly evident reductions in *Campylobacter* levels from incoming to final products shown in Figure 5, the reductions are not sufficient to eliminate the organism from particular samples to render the sample undetectable. A reduced carcass sampled for *Campylobacter* prevalence will show as positive if detection methods are used to estimate microbial levels, but the concentration of the pathogen on a per-sample basis will be significantly reduced but not to the level to be detected as negative when full enrichment is applied for detection methodologies [2]. Anecdotical information from poultry processors states that some of the interventions used in commercial plants work for *Salmonella* but not for *Campylobacter* reductions. If the performance is measured by prevalence data, the results will indicate minimal effect of the intervention scheme, but when seen through the quantification light, reductions are significant but not enough to turn samples fully negative under pathogen-detection systems based on sample enrichment.

## 4. Conclusions

The development and use of bio-mapping data, including proper data visualization, improves the tools needed for decision making in food safety for commercial poultry operations. The correct planning and management of data already collected but further analyzed allows companies to understand the microbiological aspect of their process not only in the topics mentioned in this paper but also for other approaches that can be evaluated, such as the development of statistical process control parameters, trend analysis, risk analysis, and the use of this information as historical data in validation studies. Nowadays, microbial data mining and analysis should be one of the basic components taken into account in a food safety team in food processing companies at the moment of making decisions, as data correctly used can give strong support and will permit the development of further research ideas to improve the management of their food safety process.

## Figures and Tables

**Figure 1 foods-12-00898-f001:**
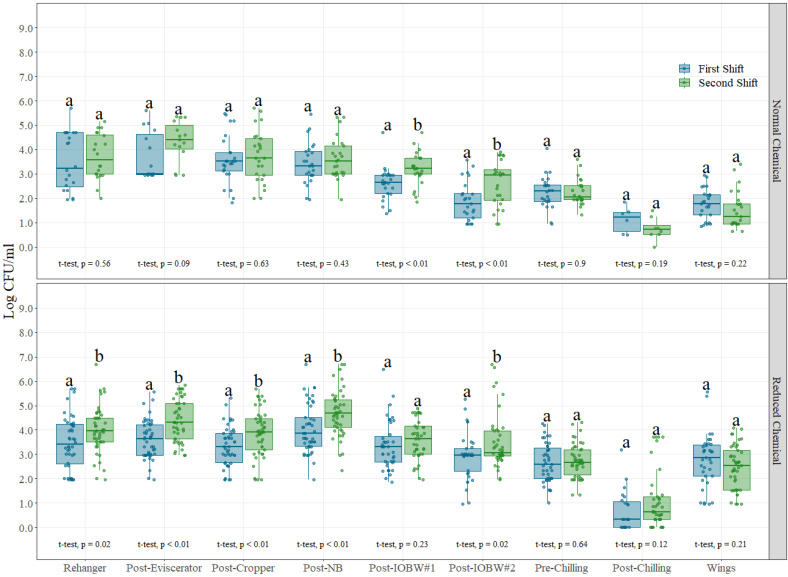
Enterobacteriaceae counts (Log CFU/mL) on each of the nine sampling locations under normal process interventions (**above**) and reduced chemical process interventions (**below**) on chicken and part rinses collected during different shifts. In each boxplot, the horizontal line crossing the box represents the median, the bottom and top of the box are the lower and upper quartiles, the vertical top line represents 1.5 times the interquartile range, and the vertical bottom line represents 1.5 times the lower interquartile range. (a,b) For normal and reduced chemical interventions at each sampling locations, boxes with different letters are significantly different according to *t*-test analysis at *p* < 0.05. The points represent the actual data points.

**Figure 2 foods-12-00898-f002:**
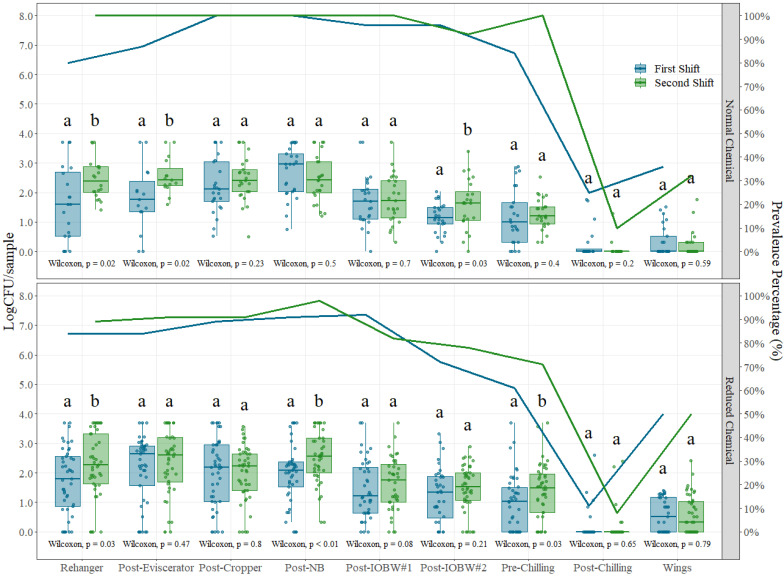
*Campylobacter* counts (Log CFU/sample) and prevalence (solid lines) on each of the nine sampling locations under normal process interventions (**above**) and reduced chemical process interventions (**below**) on chicken and part rinses collected during different shifts. In each boxplot, the horizontal line crossing the box represents the median, the bottom and top of the box are the lower and upper quartiles, the vertical top line represents 1.5 times the interquartile range, and the vertical bottom line represents 1.5 times the lower interquartile range. (a,b) For normal and reduced chemical interventions at each sampling location, boxes with different letters are significantly different according to Wilcoxon’s test analysis at *p* < 0.05. The points represent the actual data points.

**Figure 3 foods-12-00898-f003:**
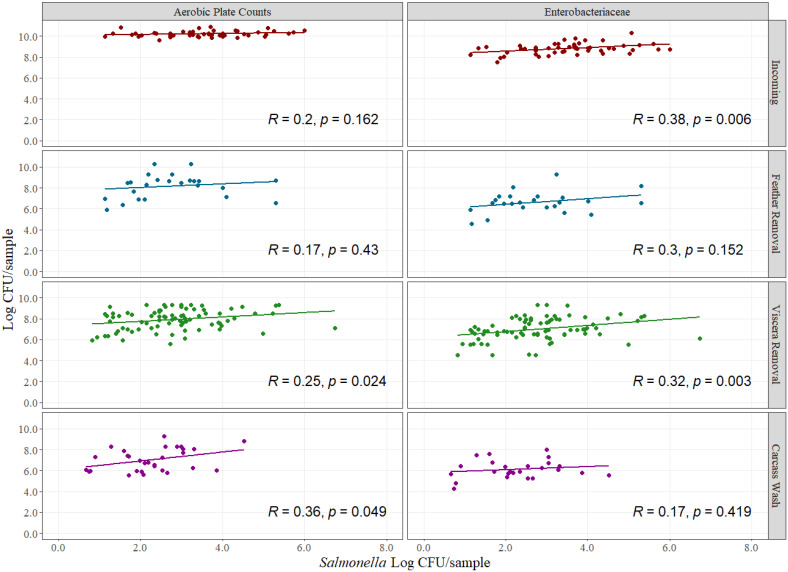
Pearson’s correlation analysis between Enterobacteriaceae (**right**) and mesophilic aerobic counts (**left**) (Log CFU/sample) with *Salmonella* counts (Log CFU/sample) at each of the four combined sampling locations on whole carcass chicken rinses. The points represent the actual data points.

**Figure 4 foods-12-00898-f004:**
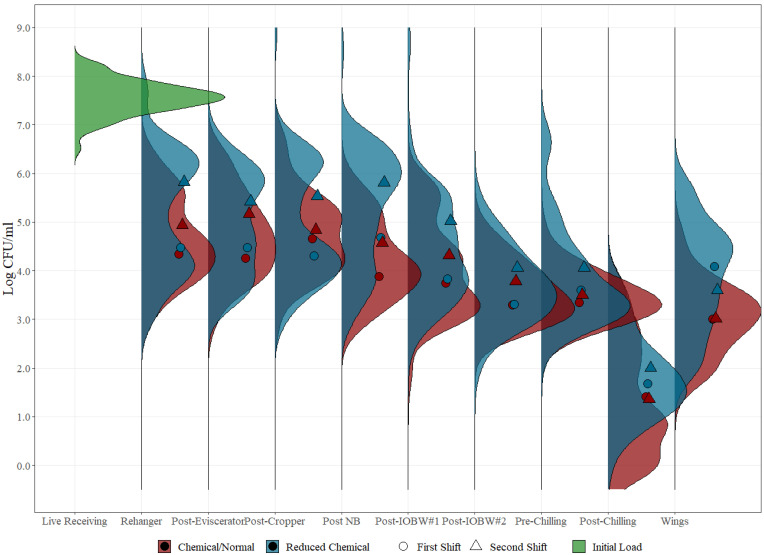
Kernel density estimation for mesophilic aerobic counts (Log CFU/mL) on each of the ten sampling locations evaluated under normal process interventions (red) and reduced chemical process (blue) intervention schemes on whole carcass chicken and part rinses. The dots represent the average value for mesophilic aerobic counts on each sampling location under both process interventions (normal and reduced) during different shifts (circle vs. triangle).

**Figure 5 foods-12-00898-f005:**
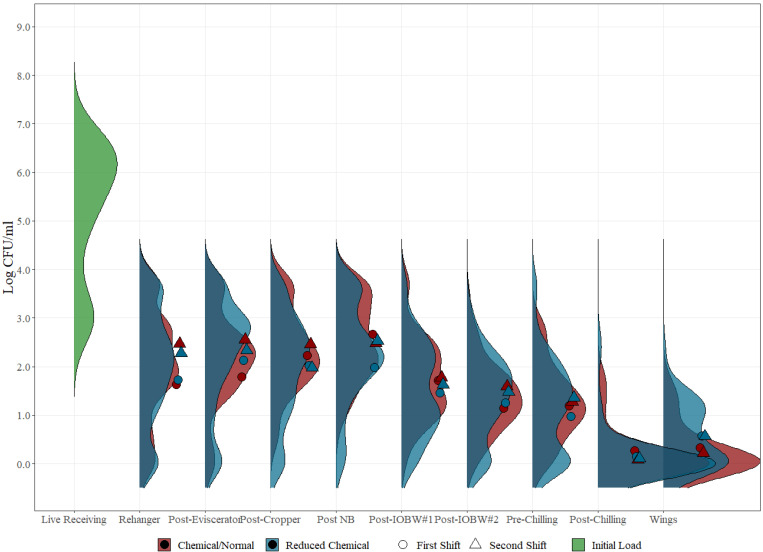
Kernel density estimation for *Campylobacter* counts (Log CFU/mL) on each of the ten sampling locations evaluated under normal process interventions (red) and reduced chemical process (blue) intervention schemes on whole carcass chicken and part rinses. The dots represent the average value for *Campylobacter* counts on each sampling location under both process interventions (normal and reduced) during different shifts (circle vs. triangle).

## Data Availability

Data available on request from the corresponding author. The data are not publicly available due to privacy from the beef processing partner that allowed the project to be conducted within their beef processing environment.

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
