# Peer review of "Data-Mining Poultry Processing Bio-Mapping Counts of Pathogens and Indicator Organisms for Food Safety Management Decision Making"

_foods, 2023, doi:10.3390/foods12040898_

Round 1

Author Response

Corrections can be found in the attached document.

Reviewer 2 Report

find attached file

Author Response

(The authors gave the same response as above.)

Reviewer 3 Report

In this manuscript, the use of Bio-mapping of bacteria was studied to see the safety in food processing.

The following comments are made:

1. The document is difficult to read since the paragraphs are very large, this makes us not appreciate the data of interest. It is recommended to divide into shorter paragraphs to improve your reading.

2. Section 2.1. You indicate that the methodology is in reference 2. However, you must briefly describe what pathogens you detected and how you did it, since if reference 2 is not reviewed, it is not possible to know what and how you did the job.

3. Line 178. “post IOBW #1, post IOBW #2”. Say what the abbreviations mean

4. Regarding the results shown in Figures 1 and 2, it can be seen that the CFU is similar in the normal reduced chemical process conditions. Discuss what explanation you give for this, since a greater amount of bacteria would be expected in reduced chemical processes. Explain how reduced the treatment is with respect to normal.

5. Lines 315. Put what EB and APC mean.

6. Line 325. “USDA-FSIS.” Put what they mean

7. Appendices A to H. You can be made in bar graphs so that the data can be better appreciated.

8. Your results show that microorganisms are not eliminated. Would you propose measures to eliminate microorganisms or only stay at the allowed CFU level? What happens with bacteria resistant to treatments? What happens to bacteria that remain persistently like Listeria? You could discuss this.

Author Response

(The authors gave the same response as above.)

Round 2

Reviewer 3 Report

The answer to suggestions 4 and 8 is that they are in the article by De Villena et al. But that manuscript is not being evaluated, it is being evaluated, so the explanations must be put in this manuscript so that it can be better understood. Since, if people do not read the manuscript of De Villena et al., before yours, they cannot fully understand it. Readers find it more tedious to search for information in previous articles. Therefore, I continue to suggest that the information from comments 4 and 8 be included in this manuscript.

Author Response

Comments are attached.
